# Full electrical manipulation of perpendicular exchange bias in ultrathin antiferromagnetic film with epitaxial strain

Jie Qi [1,6], Yunchi Zhao [2,6] ✉, Yi Zhang[2,3], Guang Yang [4], He Huang[3], Haochang Lyu[3], Bokai Shao[3], Jingyan Zhang[3], Jialiang Li[5], Tao Zhu[2,5], Guoqiang Yu [2], Hongxiang Wei[2], Shiming Zhou[1], Baogen Shen[1] & Shouguo Wang [1] ✉

Achieving effective manipulation of perpendicular exchange bias effect remains an intricate endeavor, yet it stands a significance for the evolution of ultra-high capacity and energy-efficient magnetic memory and logic devices. A persistent impediment to its practical applications is the reliance on external magnetic fields during the current-induced switching of exchange bias in perpendicularly magnetized structures. This study elucidates the achievement of a full electrical manipulation of the perpendicular exchange bias in the multilayers with an ultrathin antiferromagnetic layer. Owing to the anisotropic epitaxial strain in the 2-nm-thick $IrMn_3$ layer, the considerable exchange bias effect is clearly achieved at room temperature. Concomitantly, a specific global uncompensated magnetization manifests in the $IrMn_3$ layer, facilitating the switching of the irreversible portion of the uncompensated magnetization. Consequently, the perpendicular exchange bias can be manipulated by only applying pulsed current, notably independent of the presence of any external magnetic fields.

In the rapidly evolving field of spintronics[1-7], antiferromagnetic (AFM) materials have gained paramount importance in the development of advanced-generation memory and computing technologies[8-12], garnering extensive attention benefiting from the advantages of zero net magnetization, robustness against external perturbations, and spin dynamics at the terahertz timescale. Recent research on AFM materials has been predominantly focused on the manipulation of Néel order in bulk antiferromagnets[13-16] and the spin configuration at the antiferromagnet/ferromagnet interface, referring to the exchange bias (EB) effect[17-22]. The EB effect can induce a unidirectional exchange anisotropy, which manifests as a shift in hysteresis loops along the direction of external magnetic fields[23,24] defined quantitatively as the exchange

bias field ($H_{EB}$) together with an expanded loop. Beyond the applications in traditional spintronic devices, the manipulation of the EB effect, intricately associated with the bulk magnetic state of AFM[25], is expected to be a promising approach for detecting or writing information stored with the AFM order in advanced memory and logic devices[26,27]. Currently, two strategies can be utilized to manipulate the EB. The first one is to reorient the pinning direction via the conventional field-cooling process[24], albeit necessitating elevated temperatures and an external field, thus hindering its application in spintronic devices. The second one is electric current-driven switching[17-20,28-35]. Nevertheless, in prior studies on exchange-biased multilayers with perpendicular magnetic anisotropy (PMA), the necessity of an external

[1]Anhui Key Laboratory of Magnetic Functional Materials and Devices, School of Materials Science and Engineering, Anhui University, Hefei 230601, China. [2]Beijing National Laboratory for Condensed Matter Physics, Institute of Physics, Chinese Academy of Sciences, Beijing 100190, China. [3]Beijing Advanced Innovation Center for Materials Genome Engineering, School of Materials Science and Engineering, University of Science and Technology Beijing, Beijing 100083, China. [4]School of Integrated Circuit Science and Engineering, Beihang University, Beijing 100191, China. [5]Spallation Neutron Source Science Center, Dongguan 523803, China. [6]These authors contributed equally: Jie Qi, Yunchi Zhao. ✉e-mail: yczhao@iphy.ac.cn; sgwang@ahu.edu.cn

magnetic field for manipulating the EB is remained[17-19,29-34], posing a technological challenge in applications, and the underlying mechanism of the current switching behavior warrants in-depth exploration.

$\gamma$-IrMn$_3$, a widely adopted AFM material, has attracted significant research interest in both fundamental physics and pragmatic applications[11,12,36-38]. It is conventionally employed to pin the reference layer in information-storage devices with EB owing to its high Néel temperature ($T_N$)[39], large anisotropy[40], and resistance to corrosion[41]. For practical applications, an optimal balance between a high $T_N$ and minimized thickness of IrMn$_3$ layer becomes imperative. However, in multilayer configurations, the finite size effect results in a marked reduction in the $T_N$ of AFM layer compared to its bulk value, leading to the disappearance of the EB effect beneath a thickness threshold of the AFM layer[42,43]. Existing theories have elucidated a correlation between $T_N$ and the thickness of the IrMn$_3$ layer ($t_{IrMn}$)[44], implying that when the thickness of IrMn$_3$ is reduced to 2 nm, the $T_N$ will be suppressed to about 230 K, which is substantially below room temperature (RT). Typically, a considerable $H_{EB}$ in multilayers can be achieved when $t_{IrMn}$ exceeds 4 nm[45-48]. Therefore, one of the paramount challenges lies in manipulating the AFM exchange coupling to strike an intricate balance between the thickness and the $T_N$ of the AFM layer. Considering the sensitive dependence of exchange interaction between spins on

interatomic distance[40], the introduction of epitaxial strain emerges as a promising avenue due to its potential modulatory impact on the exchange bias effect at the antiferromagnet/ferromagnet interface[39,49-51].

In this work, a considerable perpendicular exchange bias field is achieved at RT in single-crystalline multilayers with a 2-nm-thick IrMn$_3$ layer accompanied by a distinct uncompensated magnetic structure, which is ascribed to the anisotropic epitaxial strain in the ultrathin AFM layer. Notably, the full electrical manipulation of the perpendicular exchange bias, obviating the need for external magnetic fields, is successfully realized. This research is instrumental in enhancing our understanding about the interplay of the exchange bias effect and epitaxial strain in multilayered structures and can greatly augment the roadmap for the design and deployment of versatile spintronic devices.

## Results and discussion

### Perpendicular exchange bias effect with ultrathin IrMn$_3$ layer

The epitaxial Pt (3 nm)/Co (1 nm)/$\gamma$-IrMn$_3$ (2 nm) multilayer was deposited onto an Al$_2$O$_3$ (0001) substrate as shown in Fig. 1a. The MgO (2 nm)/Cr (2 nm) was deposited as capping layers to avoid degradation. The single-crystalline structure and element distributions were

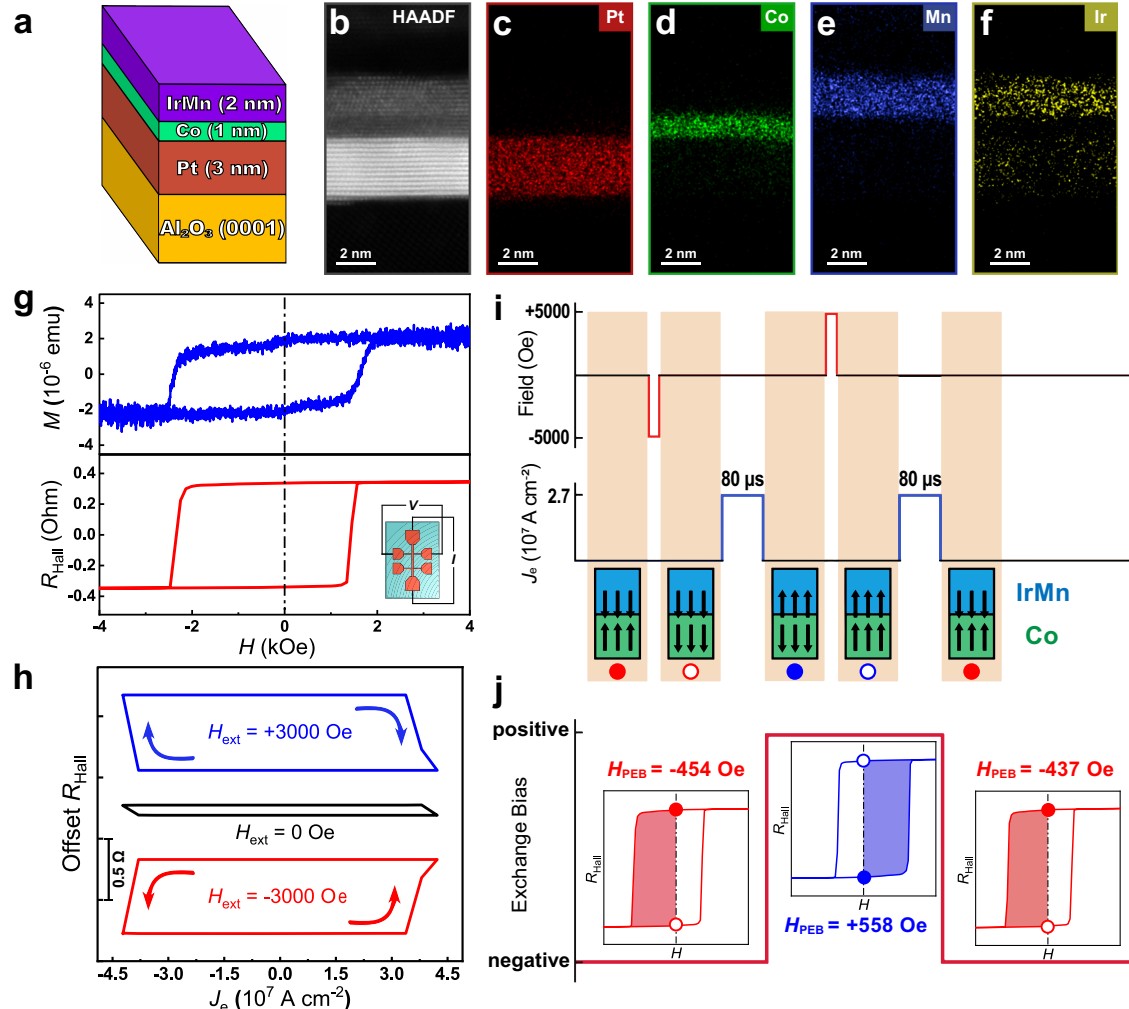

**Fig. 1 | The field-free perpendicular exchange bias switching process.**
**a** Schematic structure of the epitaxial Pt/Co/IrMn$_3$ multilayer. **b** HAADF image of the multilayer. **c**–**f** EDS mappings of Pt, Co, Mn, and Ir corresponding to the region in (**b**). **g** The $M$-$H$ loop and the AHE loop of the Pt/Co/IrMn$_3$ structure with the exchange bias effect. Inset: schematic structure of the patterned Hall bar for

electrical transport measurements. **h** The current-induced magnetization switching with the $H_{ext}$ set as −3000 Oe, 0 Oe, and +3000 Oe, respectively. **i** The operation sequence of the field-free PEBS process. Inset: four different states for the interfacial magnetic moments of the Co and IrMn$_3$ layers. **j** The switching process of the perpendicular exchange bias.

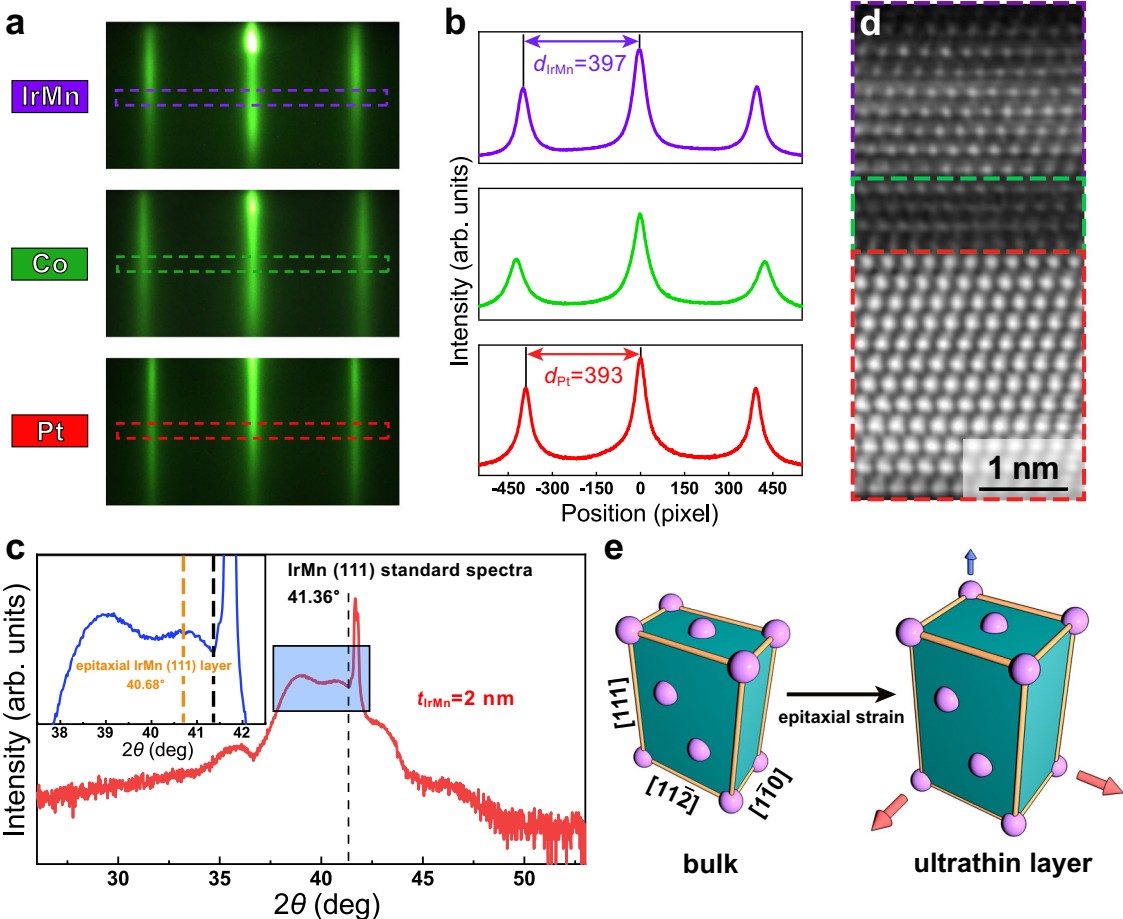

**Fig. 2 | The structural characterization of the Pt/Co/IrMn₃ multilayer. a** RHEED patterns for Pt, Co, and IrMn₃ layers, respectively, with the incident electron beam parallel to the [11$\bar{2}$0] direction of the Al₂O₃ (0001) substrate. **b** Normalized intensity gray level of the RHEED patterns of Pt, Co, and IrMn₃ layers, respectively. **c** XRD spectra of the ultrathin Pt/Co/IrMn₃ multilayer. All peaks are calibrated based on a standard powder diffraction file card of substrates for XRD with Cu-K$_\alpha$ radiation.

Inset: enlarged spectra of the selected area, where the position of the diffraction peak of the IrMn₃ layer is marked with the orange dashed line, and the peak position of the standard spectra is marked with the black dashed line. **d** High-resolution HAADF image of the Pt/Co/IrMn₃ structure. **e** Schematic atomic configurations of the bulk IrMn₃ and ultrathin IrMn₃ with anisotropic tensile strain.

characterized by cross-sectional scanning transmission electron microscopy (STEM) and energy dispersive X-ray spectroscopy (EDS). As shown in Fig. 1b, the high-angle annular dark field (HAADF) image demonstrates the high quality of the epitaxial Pt/Co/IrMn₃ multilayer with a single-crystalline structure. The discernible image contrast and EDS mappings (Fig. 1c–f) of Pt, Co, and IrMn₃ layers demonstrate their precise thickness, uniform elemental distributions, and smooth interfaces. The atomic ratio between Ir and Mn atoms in the IrMn₃ layer was evaluated to be 22.2:77.8, in good agreement with the nominal ratio of 25:75. Subsequently, the film was fabricated into Hall bar devices with a width of 15 μm utilizing conventional lithography and ion milling to perform electrical transport measurements. The anomalous Hall effect (AHE) was measured by sweeping the magnetic field parallel to the out-of-plane direction while the current was injected along the current channel as illustrated in the inset of Fig. 1g. The good PMA of the epitaxial Pt/Co/IrMn₃ structure can be confirmed by the square shape of the $R$-$H$ loop as shown in Fig. 1g, and the perpendicular exchange bias field ($H_{\text{PEB}}$) can be derived to be about 436 Oe simultaneously, in accordance with the value from the magnetization hysteresis loop obtained by vibrating sample magnetometer. It is noteworthy that such a high value of $H_{\text{PEB}}$ corresponds to a mere 2-nm-thick IrMn₃ layer (The verification of robustness and reproducibility of $H_{\text{PEB}}$ can be found in P1 of the Supporting Information), contrasting with the prevalent notion of necessitating IrMn₃ layers thicker than 4 nm to achieve

a considerable $H_{\text{PEB}}$. Additionally, the $t_{\text{IrMn}}$ dependence of the $H_{\text{PEB}}$ for the epitaxial Pt/Co/IrMn₃ multilayer was investigated (refer to P2 of the Supporting Information), which indicated a critical thickness of about 1.5 nm for the IrMn layer to achieve a perpendicular exchange bias effect at RT.

In ensuing experiments, current-induced magnetization switching (CIMS) was conducted by applying a DC pulsed current along the longitudinal channel (as detailed in the Methods Section) under an in-plane magnetic field ($H_{\text{ext}}$) of ±3000 Oe (depicted in Fig. 1h). As expected in the spin-orbit-torque (SOT) framework, the switching loops exhibited opposite switching polarities. In the absence of the in-plane magnetic field, a minimal CIMS ratio manifests, which is reasonable for $\gamma$-IrMn₃ with the 3$Q$ magnetic structure[39,52,53]. Notably, perpendicular exchange bias switching (PEBS) of epitaxial Pt/Co/IrMn₃ can be realized by applying pulsed electrical current as depicted in Fig. 1i, j. For the initial state with negative $H_{\text{PEB}}$, the magnetization of Co ($\mathbf{m}_{\text{Co}}$) is "up" which is antiparallel to the interfacial uncompensated magnetization of Mn ($\mathbf{m}_{\text{Mn}}$). Then, the $\mathbf{m}_{\text{Co}}$ can be switched to the "down" state by applying a magnetic field of −5000 Oe, while the $\mathbf{m}_{\text{Mn}}$ remains unchanged corresponding to the negative $H_{\text{PEB}}$. When a pulsed current with a density of $2.7 \times 10^7$ A cm$^{-2}$ is injected in the absence of the magnetic field, $\mathbf{m}_{\text{Mn}}$ reorients to antiparallel to $\mathbf{m}_{\text{Co}}$ in the absence of the external fields, resulting in that the direction of $H_{\text{PEB}}$ was reversed to be positive, as shown in the insets of Fig. 1j. Moreover if

the $\mathbf{m}_{Co}$ is aligned to "up" state by applying a magnetic field of +5000 Oe and followed by a pulsed current, the $H_{PEB}$ can be switched back to the initial negative state. The insets in Fig. 1i reveal the four different states of the magnetic moment configurations at the interface during the PEBS process. The experimental results indicate that the sign of the perpendicular exchange bias can be switched by injecting the pulsed current based on a defined magnetization state of Co. In contrast to previous studies on perpendicular exchange bias, such current-driven manipulation can be realized without an external magnetic field. The magnetic field herein solely serves to change the magnetization direction of the Co layer.

### Structural characterization and epitaxial strain
Comprehensive characterizations were performed to give an understanding of the crystalline structure of epaxial Pt/Co/IrMn$_3$ multilayers. Figure 2a shows the in situ reflection high-energy electron diffraction (RHEED) patterns with sharp and continuous diffraction streaks corresponding to the Pt, Co, and IrMn$_3$ layers, respectively, indicative of the formation of a high-quality epitaxial structure. The distances between the reflection streak and the first-order diffraction streak ($d$) of Pt ($d_{Pt}$) and IrMn$_3$ ($d_{IrMn}$) layers can be quantitatively analyzed through the line scan intensity curves of the RHEED patterns as shown in Fig. 2b, which are derived as a ratio of $d_{IrMn}/d_{Pt} = 1.01$. It should be noticed that the lattice parameter of the Pt layer is close to the bulk value, attributed to the negligible lattice mismatch (0.95%) existing between the Pt (111) and the Al$_2$O$_3$ (0001) lattices and the epitaxial-strain release through a high-temperature annealing process. Based on the bulk lattice constant of Pt (3.923 Å), the lattice constant of the ultrathin IrMn$_3$ is extracted as ~3.88 Å, representing a 2.7% expansion relative to its bulk value (3.778 Å). The distortion in the ultrathin IrMn$_3$ layer elucidated by the RHEED pattern signifies in-plane tensile strain due to a large lattice mismatch between Co and IrMn$_3$ (6.57%).

Figure 2c presents the X-ray diffraction (XRD) patterns of the Pt (3)/Co (1)/IrMn$_3$ (2) (in nm). An enlarged view of the selected region in the inset highlights the (111) diffraction peak of 2-nm-thick IrMn$_3$ (marked by the orange dashed line), which is located at $2\theta = 40.68°$. The peak's position, to the left of the bulk value derived from the standard spectra of IrMn$_3$ (41.36°, marked by the black dashed line), implies augmented interatomic distances along [111] direction. This corroborates the presence of out-of-plane tensile strain in the ultrathin IrMn$_3$ layer, quantitatively extracted to be 1.35% according to Bragg's law.

Moreover, the epitaxial strain is directly ascertainable based on the high-resolution HAADF image, revealing a well-defined atomically layered structure as shown in Fig. 2d. The average in-plane (along [11$\bar{2}$] direction) and out-of-plane (along [111] direction) atomic interlayer distances are derived to be 2.42 Å and 2.23 Å, respectively. In comparison with bulk IrMn$_3$ (atomic interlayer distances of 2.31 Å and 2.18 Å along [11$\bar{2}$] and [111] directions, respectively), there exists a tensile strain of about 2.3% along [111] direction and 4.8% along [11$\bar{2}$] direction. The in-plane tensile strain is larger than that along the out-of-plane direction, consistent with the RHEED patterns and the XRD results. Figure 2e shows the schematic atomic arrangements characterizing the crystalline structure of both bulk and ultrathin IrMn$_3$. Within the ultrathin IrMn$_3$ layer, an anisotropic lattice distortion is evident, where the magnitude and direction of this distortion clearly are annotated with arrows.

### The specific global uncompensated magnetization within the IrMn$_3$ layer
It is widely acknowledged that the exchange bias effect in multilayers is intricately linked to the thickness of the AFM layer[42,43]. Based on the study by L. Frangou et al., which accounts for magnetic phase transitions and finite size-scaling[54], a linear relationship between $T_N$ and the thickness of the AFM layer ($t_{IrMn}$) has been discerned[44]. Specifically, for

$t_{IrMn} < n_0$ (the finite divergence of the spin-spin correlation length near the $T_N$, $n_0 \approx 2.7$ nm for IrMn$_3$[44]), $T_N(t_{IrMn}) = T_N(\text{bulk})\frac{t_{IrMn}-d}{2n_0}$, wherein $T_N$ (bulk) represents the Néel temperature of the IrMn$_3$ bulk (700 K[55]), and $d$ is the interatomic distance, equating to 0.22 nm for IrMn$_3$ along [111] direction. With the reduction of $t_{IrMn}$ to 2 nm, the $T_N$ of IrMn$_3$ will be suppressed below RT (about 230 K). Typically, $t_{IrMn}$ needs to be larger than 4 nm to obtain a considerable $H_{EB}$ in multilayers.

On the other hand, the significantly diminished volume anisotropy energy ($E_A$) of the 2 nm IrMn$_3$ layer proves insufficient to resist the exchange energy at the antiferromagnet/ferromagnet interface ($E_{int}$)[45]. As the $\mathbf{m}_{Co}$ rotates with the magnetic field, the $\mathbf{m}_{Mn}$ will be switched simultaneously, only resulting in an increase in coercivity. When the IrMn layer is thicker, there will be enough energy to produce exchange bias accompanied by an increased coercivity. This discussion, however, overlooks the influence of crystalline structure on EB[49,50]. Factoring in the nonnegligible tensile strain in the ultrathin IrMn$_3$ layer, the contribution from the aforementioned anisotropic epitaxial strain becomes imperative to decipher the abnormal $H_{PEB}$ of 2-nm-thick IrMn$_3$ in this work. More experimental evidence was gleaned from two control samples with distinct strain to further verify its effect on EB[56,57] (refer to P3 of the Supporting Information). The results reveal that augmented epitaxial strain directly corresponds to a more significant exchange bias field, underscoring the pivotal role of lattice strain in modulating the exchange bias effect.

Furthermore, the atomistic simulation by the VAMPIRE software package was performed to further clarify the physical mechanism for the influence of the epitaxial strain in the single-crystalline IrMn$_3$[58,59]. A system with Co (1 nm)/IrMn$_3$ (2 nm) was constructed in fcc-(111) orientation to duplicate the experimental structure as depicted in Fig. 3a. The effective exchange interactions between Mn atoms were limited to nearest ($-3.7 \times 10^{-21}$ J/link) and next-nearest ($3.0 \times 10^{-21}$ J/link) neighbors[38,60,61]. Other detailed parameters can be found in the P4 of the Supporting Information. For the configuration without epitaxial strain, no obvious $H_{PEB}$ manifests (refer to Fig. S9 of the Supporting Information). Factoring in the anisotropic tensile strain in ultrathin IrMn$_3$, an additional magnetocrystalline anisotropy (MCA) associated with an extra anisotropy energy ($E_{MCA} = 1.0 \times 10^{-22}$ J/atom) was introduced in the atomistic simulations. Upon applying a zero-field cooling process from 600 K, with $\mathbf{m}_{Co}$ locked in either the "up" or "down" states, the resulting simulated hysteresis loop at 300 K manifests a distinctive $H_{PEB}$ of −413 Oe (red curve) or 377 Oe (blue curve). This suggests that the $E_{int}$ can be overcome by introducing $E_{MCA}$, resulting in the exchange bias field of the Pt/Co/IrMn$_3$ with epitaxial strain. Additionally, the simulated PEBS process by the zero-field cooling process is in accordance with the experimental results of the field-free switching behavior by pulsed current, revealing a nonnegligible effect of the thermal accumulation within the device (refer to P6 of the Supporting Information).

More specifically, the interfacial net magnetization of the IrMn$_3$ layer with $E_{MCA} = 1.0 \times 10^{-22}$ J/atom was analyzed with the hysteresis loops to elucidate the origin of the exchange bias effect. According to previous research, the uncompensated magnet moment of IrMn$_3$ consists of a reversible moment ($\mathbf{n}_{re}$) and an irreversible moment ($\mathbf{n}_{ir}$)[38], which correspond to the reversible and irreversible spins of the Mn atoms, respectively. The $\mathbf{n}_{ir}$ is strongly coupled to bulk AFM structure, which cannot rotate with the magnetic field, facilitating a pinning effect on the ferromagnetic (FM) layer. While the $\mathbf{n}_{re}$ can be switched with $\mathbf{m}_{Co}$, augmenting the coercivity of the system. The directions of $\mathbf{n}_{re}$ and $\mathbf{n}_{ir}$ corresponding to the opposite exchange bias effect can be determined by their components along different axes as depicted in Fig. 3b. At the positive (negative) saturation state, the interfacial net magnetization of the IrMn$_3$ layer, which is defined as $\mathbf{n}_{Pos}$ ($\mathbf{n}_{Neg}$), can be expressed as $\mathbf{n}_{Pos} = \mathbf{n}_{ir} + \mathbf{n}_{re}$ ($\mathbf{n}_{Neg} = \mathbf{n}_{ir} - \mathbf{n}_{re}$). The magnitude of $\mathbf{n}_{ir}$ can be meticulously quantified by the vertical offset of the hysteresis loop, which is equal to ($\mathbf{n}_{Pos} + \mathbf{n}_{Neg}$)/2, wherein the value of

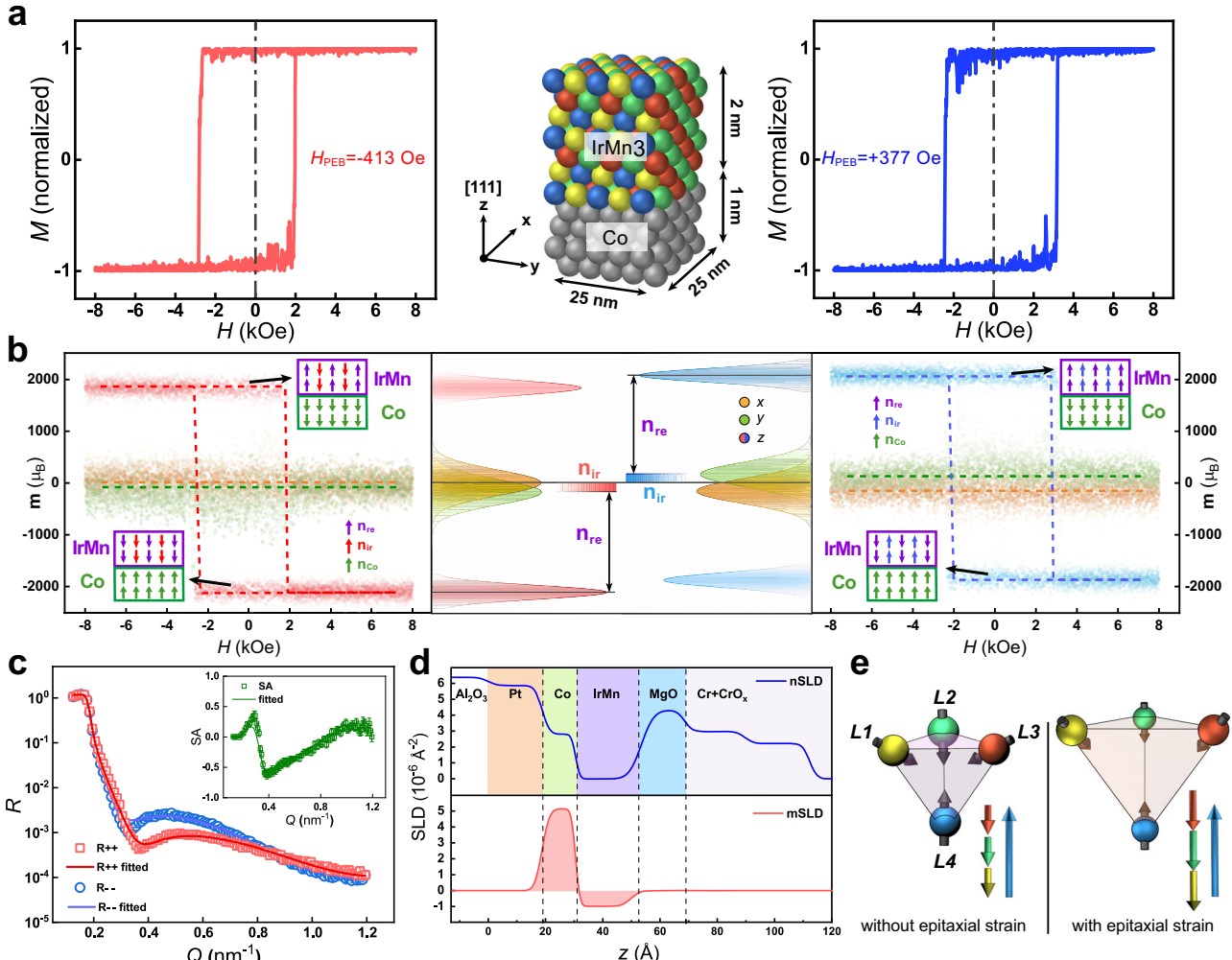

**Fig. 3 | Atomistic simulation and the polarized neutron reflectometry.**
**a** Schematic representation of the Co/IrMn₃ bilayer and the corresponding simu-lated hysteresis loops at 300 K with the $E_{MCA} = 1.0 \times 10^{-22}$ J/atom. The red (blue) curve corresponds to the negative (positive) exchange-biased state. **b** The com-ponents of the net magnetic moment switching behaviors of the strained IrMn₃ layer with negative (left) and positive (right) exchange bias effect. The middle part depicts the statistical distribution of the magnetic moments along each axis. Insets: the schematic magnetization configuration of the IrMn₃ and the Co layers at negatively and positively saturated magnetization states. **c** Reflectivity ($R$) of the

sample as a function of wave factor ($Q$), which was performed under a 1.6 T in-plane magnetic field at RT. Open symbols and solid lines represent the experimental data and theoretical fits, respectively. Inset: the SA curves with error bars representing one standard deviation. **d** The nSLD and mSLD depth profiles of the sample. An in-plane field of 16,000 Oe was applied during PNR measurements. **e** Typical 3$Q$ spin structure with 109.5° between spins of γ-IrMn₃ (left) and the unique uncompen-sated magnetic structure of IrMn₃ with anisotropic tensile strain (right). The arrows in the insets represent the components along the (111) direction of the four mag-netic sublattices.

($\mathbf{n}_{Pos} - \mathbf{n}_{Neg}$)/2 is derived as the magnitude of $\mathbf{n}_{re}$. The $\mathbf{n}_{re}$ and $\mathbf{n}_{ir}$ are presented with the statistical distribution of the magnetic moments along each axis as shown in the middle portion of Fig. 3b, which are extracted from the dispersive hysteresis loops attributed to the ther-mal disturbance of spins in the atomistic simulation[62]. Notably, the $\mathbf{n}_{re}$ predominantly aligns with the $z$-axis for both states with negative (left portion) and positive (right portion) $H_{PEB}$. More precisely, the $\mathbf{n}_{re}$ flips with the $\mathbf{m}_{Co}$ retaining AFM coupling, while the $\mathbf{n}_{ir}$ remains fixed as illustrated in the insets. The values of the $\mathbf{n}_{ir}$ for the two loops are extracted to be $-130\,\mu_B$ (for the negative $H_{PEB}$) and $107\,\mu_B$ (for the positive $H_{PEB}$), respectively, which reverse in accordance with the $\mathbf{m}_{Co}$ during the zero-field cooling process, thereby engendering oppo-site $H_{PEB}$.

To elucidate the detailed AFM structure of the ultrathin IrMn₃ layer, the spatially resolved magnetization along the depth of the Pt (3 nm)/Co (1 nm)/IrMn₃ (2 nm) film was explored by polarized neutron reflectometry (PNR) measurements. The details of the nuclear profile and the magnetization configuration can be determined by the non-

spin-flip reflectivities $R_{++}$ (spin-up neutrons) and $R_{--}$ (spin-down neu-trons) and the corresponding spin-asymmetry ratios ($R_{++} - R_{--}$) / ($R_{++} + R_{--}$) (SA) with respect to wave vector transfer ($Q$) as shown in Fig. 3c. The corresponding profiles of the nuclear (nSLD) and magnetic scattering-length density (mSLD) are shown in Fig. 3d. Upon analysis, the thickness of the IrMn₃ layer is detected to be 21.6 Å, corroborated by the nSLD profiles, consistent with the nominal thickness. Notably, there exists an abnormality for the ultrathin IrMn₃ layer in terms of the mSLD profile. In conventional scenarios, uncompensated magnetiza-tion in the IrMn₃ layer without epitaxial strain can be detected pre-dominantly nearby the ferromagnet/IrMn₃ interface, which is typified by AFM coupling with the FM layer[17]. Contrarily, the 2-nm-thick IrMn₃ possesses pronounced uncompensated magnetization in the full range of the AFM layer exhibiting AFM coupling with Co. Previous neutron-scattering experiments indicate that the chemically disordered IrMn₃ with tetrahedral (3$Q$) spin structure possesses compensated magne-tization with 109.5° between spins of the four magnetic sublattices[39]. Specifically, the anisotropic tensile strain along in-plane and out-of-

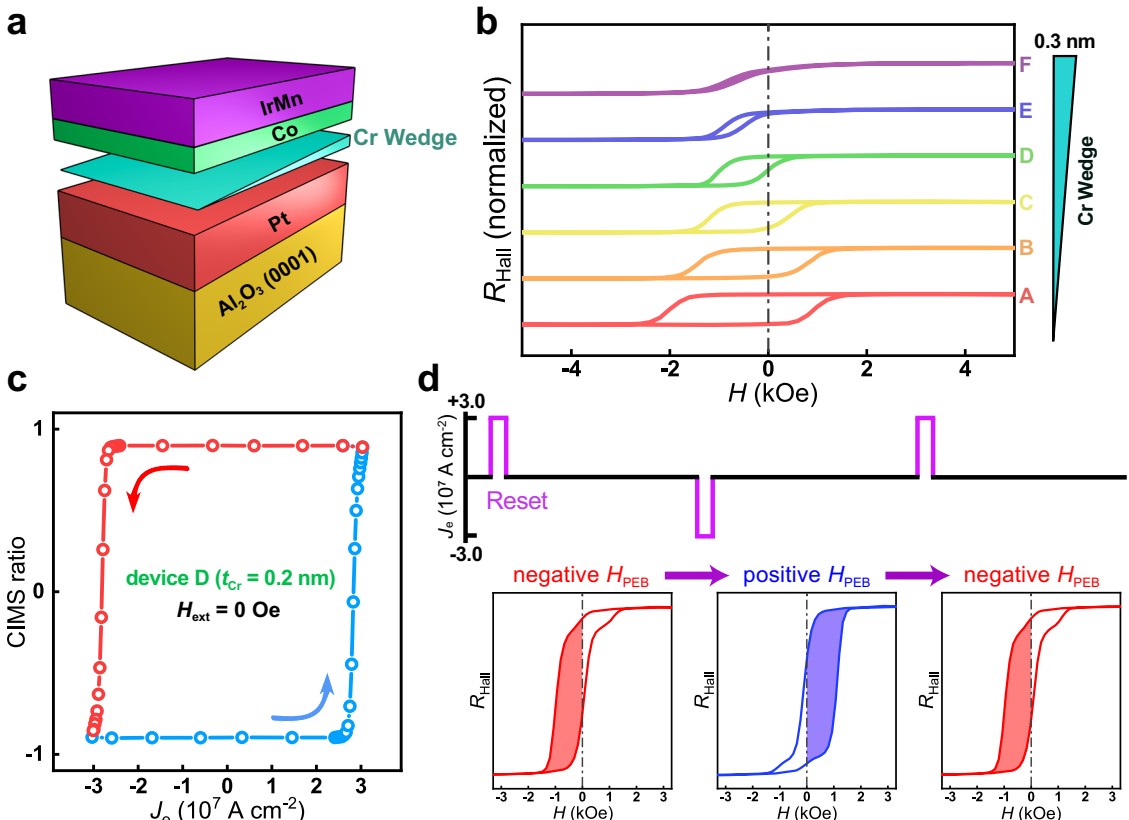

**Fig. 4 | Full electrical manipulation of the PEBS process. a** Schematic structure of the epitaxial multilayer with a Cr wedge. **b** AHE loops of six devices along the wedge direction corresponding to different thicknesses of the Cr insertion. **c** SOT-induced field-free magnetization switching behavior of the device D. **d** The operation sequence of the full electrical manipulation of the PEBS process.

plane directions may lead to a decrement of the exchange coupling among Mn atoms, leading to a diminished in-plane exchange energy compared to its out-of-plane counterpart. This results in the inability of the ultrathin $IrMn_3$ to retain its typical $3Q$ compensated magnetic structure (as depicted in the left of Fig. 3e). The attenuated AFM coupling among magnetic sublattices L1, L2, and L3 leads to an angular deviation of the constituent spins away from the in-plane direction, engendering a net magnetization in $IrMn_3$ as illustrated in the right of Fig. 3e. In the configuration involving a 2-nm-thick single-crystalline $IrMn_3$ layer, the residual epitaxial strain remains in such ultra-thin thickness. Consequently, a reversible net magnetization, attributable to this unreleased epitaxial strain, is detectable spanning the entire AFM layer, and this is validated by the PNR results.

As aforementioned, the $n_{ir}$ is robustly pinned by the bulk AFM structure, precluding its synchronous switching with the $m_{Co}$, thereby culminating in the manifestation of the exchange bias effect. However, when considering the ultrathin $IrMn_3$ layer with globally uncompensated magnetic moments, the $n_{re}$ surrounding $n_{ir}$ within the AFM $IrMn_3$ layer can be switched with $m_{Co}$. Consequently, $n_{ir}$ transforms to a metastable state with higher energy, manifesting as a reduced energy barrier for switching. During the PEBS process, the increase in temperature resulting from the pulsed current leads to a reduction of $E_A$, which weakens the coupling between $n_{ir}$ and bulk spins within the $IrMn_3$ layer. Specifically, when $E_A$ falls below $E_{int}$, the $n_{re}$ maintains stability owing to the coupling with Co, while the $n_{ir}$ becomes decoupled from the bulk spins due to the diminished $E_A$. It thus can be speculated that the $n_{ir}$ prefers alignment with $m_{Co}$ (thus $n_{re}$) in terms of energy, rather than with the bulk spins of $IrMn_3$. As a result, throughout the PEBS process, the orientation of $n_{ir}$ is exclusively influenced by $m_{Co}$. This potentially allows the $n_{ir}$ to be reoriented and recoupled with the $IrMn_3$ bulk spins to revert to the lowest energy

state, thereby facilitating the field-free switching of the perpendicular exchange bias.

## Full electrical manipulation of PEBS

Building upon the underlying mechanism of PEBS driven by current, an innovative strategy to achieve full electrical manipulation entails the conception of a sample with the feasibility of field-free SOT switching of Co magnetization. Therefore, another structure was constructed consisting of Pt (3)/Cr wedge (0 ~ 0.3)/Co (1)/$IrMn_3$ (2) (thickness in nm) as illustrated in Fig. 4a. An array of Hall bar devices was fabricated with the current channel perpendicular to the Cr wedge direction. The PMA and the $H_{PEB}$ can be simultaneously investigated by the AHE measurements. Figure 4b presents the AHE loops of six devices along the wedge direction. The results indicate that the PMA is sensitive to the thickness of the Cr insertion, and the exchange bias effect is maintained for all the devices. Figure 4c depicts the current switching curve of device D corresponding to a 0.2-nm Cr insertion. The device exhibits the viability of field-free magnetization switching driven by SOT owing to the lateral symmetry breaking, which originates from an in-plane component of the exchange bias field. The detailed discussion can be found in P7 of the Supporting Information[63,64]. Based on this, the full electrical manipulation of the PEBS process can be realized as exhibited in Fig. 4d. By applying a negative pulsed current with a density of $-3 \times 10^7 A\,cm^{-2}$ without external magnetic fields, the $H_{PEB}$ of the device switches from negative ($-424$ Oe) to positive (488 Oe) state. Similarly, a positive pulsed current with a density of $3 \times 10^7 A\,cm^{-2}$ can also lead to the full electrical manipulation of the PEBS process from the positive (488 Oe) back to the negative ($-403$ Oe) state. Additionally, the intermediary state of the PEBS process exhibiting double-biased switching behavior can be manipulated by tuning the density of the pulsed current, which is discussed in P8 of the Supporting

Information. This method of full electrical manipulation of PEBS holds significant implications for optimizing spintronic device configurations, and promotes the further application of the correlative devices.

In summary, we have reported a considerable exchange bias effect in a single-crystalline multilayered structure with a 2-nm-thick IrMn$_3$ layer. The anisotropic epitaxial strain within the ultrathin AFM layer engenders additional magneto-crystalline anisotropy energy, which results in the exchange bias effect, together with a specific global uncompensated magnetization of IrMn$_3$ corroborated by PNR. Noteworthy, by introducing a wedged insertion at the Pt/Co interface, the full electrical manipulation of perpendicular exchange bias is successfully realized. This study will shed light on the physical intention of the correlation between epitaxial strain and exchange bias effect in multilayered structures. It also seeks to pave a feasible pathway for establishing a practical route for the harnessing of the PEBS process, thereby bearing significant implications for the development and optimization of future spintronic memory and logic devices.

## Methods
### Film deposition and device fabrication
The epitaxial films were deposited onto polished Al$_2$O$_3$ (0001) single-crystal substrates, MgO (111) single-crystal substrates, and thermally oxidized Si substrates using a molecular beam epitaxy (MBE) system with a base pressure superior to $7.0 \times 10^{-11}$ mbar. The Al$_2$O$_3$ substrates were cleaned by acetone and isopropanol in an ultrasound bath and annealed at 750 °C for 2 h in the MBE chamber prior to the deposition. A 3-nm Pt layer was directly deposited on the surface of the Al$_2$O$_3$ and MgO substrates at 100 °C and was annealed for half an hour at 600 °C to release the lattice stress and to smooth the surface. Afterward, a 1-nm Co layer and 2-nm IrMn$_3$ layer were epitaxially deposited at 100 °C under an out-of-plane magnetic field (about 350 Oe). No post-annealing process was performed to avoid diffusion of Co and IrMn$_3$ and simultaneously maintain the lattice strain in the IrMn$_3$ layer. The polycrystalline film discussed in P3 of the Supporting Information was deposited onto Si substrates with a 500 nm thermal oxide layer using the same MBE system. All the layers were deposited at RT and the Pt layer was annealed at 350 °C. Finally, 2-nm MgO and 2-nm Cr capping layers were subsequently deposited at RT as the capping layers to avoid the degradation of the IrMn$_3$ layer. For the sample prepared for the PNR measurements, MgO (2 nm)/Cr (5 nm) capping layers were deposited to amplify the signal-to-noise ratio in the PNR measurement and to improve the precision of data fitting with an increased thickness. The deposition rates for Pt, Co, Ir, and Mn were controlled to be 0.040, 0.020, 0.020, and 0.054 Å/s, respectively, which were well calibrated by in situ quartz crystal microbalance before deposition. The wedged Cr insertion was deposited with a rate of 0.020 Å/s under a removing shutter. After deposition, the films were patterned into Hall bar devices using standard UV lithography and etching processes. The width of the Hall bar is 15 μm.

### Structural characterization
The formation of epitaxial (textured) structure and the high-quality surface was identified by RHEED and XRD with Cu-K$_\alpha$ radiation ($\lambda = 1.542$ Å). STEM and EDS characterizations were performed using a spherical-aberration-corrected FEI Themis Z microscope.

### Electrical transport measurement
The exchange bias effective field at RT was determined by the electric transport measurements, carried out at several home-built magneto-electric transport measurement systems, including electromagnets, Keithley 2604B current source, and Keithley 2182A voltage meter. The electric transport measurements at different temperatures were carried out using a physical property measurement system (PPMS Dyna-Cool-14T). For the CIMS measurements, a pulsed DC electrical current

was applied with a duration of 150 μs by Keithley 6221 current source. After an interval of 3 s for each data point, the Hall voltage was recorded under a small DC excitation current (500 μA). During the measurements of electrical manipulation for the exchange bias, a pulsed DC electrical current with a duration of 80 μs was applied by Keithley 6221 current source.

### Polarized neutron reflectometry measurements
The PNR measurements were performed using the Multipurpose Reflectometer at the China Spallation Neutron Source. The experiments were performed under a 1.6 T in-plane magnetic field at RT. The neutron reflectivity curves were recorded as a function of the momentum transfer $Q$. Neutron beam was impinged onto the sample with an incident angle and was collected by a $^3$He detector. The collected data were fitted using the *GenX* software package.

### Atomistic simulation
Spin dynamics simulations were performed solving the stochastic Landau–Lifshitz–Gilbert equation with a Heun numerical scheme. A system with Co (1 nm)/IrMn$_3$ (2 nm) containing 147,392 atoms in total was constructed in (111)-oriented with fcc crystalline structure to reproduce the structure in experiments.

## Data availability
The datasets generated during and/or analyzed during the current study are available from J.Q. (j.qi@ahu.edu.cn), and are provided in the Source Data file.

## Code availability
There is no mathematical algorithm or custom code that is deemed central to the conclusion of this manuscript. However, the custom codes that were simply used for data analysis are available from J.Q. (j.qi@ahu.edu.cn) upon reasonable request.

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

## Acknowledgements

This work was supported by the National Key Research and Development Program of China (Grant 2022YFA1402602), the Natural Science Foundation of China (Grants 52130103, 12104486, 51971026, 12374099, 12174426, 52201200, 52201288), and the Beijing Natural Science Foundation Key Program (Grant Z190007).

## Author contributions

J.Q. and Y.C.Z. contributed equally to this work as the first authors. S.G.W. initialized, conceived, and supervised the project. J.Q. prepared the samples, fabricated devices, and performed measurements with contributions from Y.C.Z., Y.Z., G.Y., H.C.L., and B.K.S. Y.Z. and H.H. performed the atomistic simulations. T.Z. and J.L.L. performed and analyzed the neutron reflectometry. J.Q. and Y.C.Z. wrote the paper with the help of J.Y.Z., G.Q.Y., H.X.W., S.M.Z., B.G.S., and S.G.W. All authors discussed the results and commented on the manuscript.

## Competing interests

The authors declare no competing interests.
