## [Peer Review File · Nature Communications]

Full electrical manipulation of perpendicular exchange bias in ultrathin antiferromagnetic film with epitaxial strainREVIEWER COMMENTS

Reviewer #1 (Remarks to the Author):

This paper presents a study of perpendicular exchange bias and its electrical control in strained IrMn₃/Co heterostructures grown epitaxially on a single-crystal Al₂O₃ (0001) wafer. The authors achieve a large room temperature exchange bias in very thin (2 nm) IrMn₃ based heterostructures. This is a surprising results given that such small thickness is assumed to reduce Neel temperature below room temperature due to the smaller magnetic volume. The authors interpret this as a result of the epitaxial strain, and they show that the exchange bias can be switched by electric currents. Despite the potential interest of these results to the spintronics research community, the quality and expected impact of these results do not rise to the level of Nature Communications.

1. It is not clear from the introduction what the motivation is for electrical switching of perpendicular exchange bias. I assume, although this is not clearly stated by the authors, they wish to use the antiferromagnetic order to store information in a memory element and use the exchange bias to detect the antiferromagnetic Neel vector direction. If so, they need to present the case and significance of the idea more clearly. They should also cite related work that has done readout of electrically switched antiferromagnetic IrMn₃ via exchange bias to a magnetic tunnel junction, which was recently reported in in-plane MTJs and is relevant to this work. [Du et al., Nature Electronics 6, 425 (2023)]

2. IrMn₃ switching by spin-orbit torque has previously been extensively studied. In addition to the above paper, the authors should also cite and compare their results to this prior work which performs anisotropic magnetoresistance detection. [Arpaci et al., Nature Communications 12, 3828 (2021)]

3. IrMn₃ is also expected to exhibit anomalous Hall effect in the absence of a ferromagnetic Co layer. The authors should check this by growing Pt/IrMn₃ bilayers without Co for comparison, and check if the AHE and its electrical switching are also observed in this case.

4. The magnitude of the exchange bias in this experiment is surprisingly large, as noted by authors. The paper should provide more information on the number of measured samples and the statistics of the exchange bias, to evaluate the robustness and reproducibility of the result.

5. The authors should also compare to different thicknesses of IrMn₃. For instance, 3 to 4 nm IrMn₃ should have a Neel temperature above room temperature even without the epitaxial strain. Can they also observe perpendicular exchange bias and switch it by current in that case? How large is the exchange bias?

Reviewer #2 (Remarks to the Author):

A very nice study of an exchanged bias system that can be manipulated with electrical currents. By growing Co/IrMn₃ films exchange bias, even with a very thin IrMn₃ layer, has been demonstrated. The exchange bias results from the enhanced anisotropy in the film due to the tensile stress. The films are well characterised and manipulations of the moments with a combination of field and current is demonstrated. This is explained using calculations using

atomistic simulation package known as Vampire. Finally it is demonstrated how the film can be manipulated with electrical currents alone with no setting required by the magnetic field.

The noteworthy results are the demonstration of exchange bias from such a thin IrMn₃ layer. The most significant find is the manipulation of the exchange bias direction with the current.

The work is very significant to the field. A key goal of the spintronics community is manipulation of magnetism without magnetic fields e.g. electric currents or electric fields. It compares very favourably with other works.

The work supports the conclusions reasonably well but some clarifications are needed. (See below)

There are a few problems I think that need to be addressed prior to publication. I have marked up a manuscript that I will attach but here are my main points:

- 1) Line 139 onwards: you talk about the RHEED analysis from different elements in the film. I am assuming that you could distinguish between the different elements in the film by performing the RHEED at different parts of the growth process. This should be made a bit clearer.
- 2) Line 189/190: you state that as the Mco rotates around the MMn rotates with it increasing the coercivity. This is completely true but for clarity it should be stated that there would be no exchange bias which would align better with the rest of the description. This is implied before this statement but it should be made clear.
- 3) Paragraph starting at line 239: there is an explanation of the film Pt/Co/IrMn₃ and then there is reference to Fig. 3. However, in the in Fig 3d there are two more layers MgO/CrO that are not mentioned in the text. Further explanation is needed here.
- 4) PNR Results: Perhaps I have misunderstood something here but polarised neutron reflectometry would not be sensitive to the changes in magnetisation perpendicular to the interfaces in the film. Since your films are mostly magnetised in the perpendicular direction I am not sure how your analysis can work. Perhaps there is sufficient component of magnetisation parallel to the film for this to work. Could you please comment on this please.
- 5) The most important part of this work is the electric manipulation of the exchange bias. Full electrical current control is done by inserting a Cr layer. The ideal thickness is 0.2nm which is beautifully demonstrated with a wedge going along several devices. There doesn't seem to be any explanation of how this works. Why did you insert the Cr layer? Why Cr? Would this work with other elements? Further explanation is required here.

The methodology is mostly sound and well done (but see comments above)

There is mostly enough detail for the work to be reproduced. The work is well explained with relevant details.

In summary I think the work is important and definitely worthy of publication after addressing the comments 1 to 5 above (with some minor details in the marked up manuscript I have attached). It is very relevant to the field of spintronics and quantum materials.

Reviewer #3 (Remarks to the Author):

The manipulation of the perpendicular exchange bias (PEB) effect in multilayers holds

significant promise for spintronic devices, particularly in the realm of energy-efficient magnetic memory and logic devices. This study reports a considerable perpendicular exchange bias effect at room temperature in a heterostructure featuring a 2-nm-thick IrMn layer. The authors also realize a full electrical manipulation of PEB, eliminating the reliance on external magnetic fields during the switching process, which is instrumental for the application of antiferromagnetic materials in novel spintronic devices. The sample in this study is high quality and well characterized, which is very helpful to comprehensively understand the crystalline structure in the ultrathin IrMn layer. This makes the physical correlation between the epitaxial strain and PEB more convincing. Based on the above reasons, I would like to conclude that the manuscript is suitable for publication in Nature Communications. Some issues should be well clarified before publication:

1. The authors acknowledge the nonnegligible thermal effect during the switching process of the PEB. However, the SOT-induced exchange bias switching behavior also exists in systems such as IrMn/CoFeB, IrMn/CoTb, IrMn/NiFe et al. Is the spin current generated from the Pt layer or the IrMn layer important to the perpendicular exchange bias switching process in this work?
2. Figure 1g demonstrates a perpendicular exchange bias field of 436 Oe with a 2 nm-thick IrMn layer. Is this thickness critical for the observed exchange bias effect? Can it be further reduced to a thinner thickness?
3. An obvious distinction exists in the switching process of the PEB between Fig. S5 and Fig. S11. It involves an overall shift of the AHE loops towards the negative exchange bias state in Fig. S5, but a double-biased state in Fig. S11. What causes this difference?
4. According to the EDS mapping (Fig. 1f), Iridium seems to exist in both the IrMn layer and the Pt layer. Is this caused by elemental diffusion or other factors?
5. Figure 4 presents a series of devices with different Cr thicknesses. Can the full electrical manipulation of the exchange bias process be achieved in these other devices?
6. Some of the important annotations in certain figures are small, hindering readers' understanding of the results, such as Fig. 3b. It is recommended to adjust for better visibility and comprehension.

Response Letter

We are grateful to all the referees for their constructive, helpful comments and suggestions. Accordingly, we have carried out more experiments and analyses and have revised our manuscript and Supplementary Information. We believe these revisions have made our work more compelling. Below we provide point-by-point responses to the comments of the referees together with a summary of changes at last.

Reviewer #1

Comment 1: It is not clear from the introduction what the motivation is for electrical switching of perpendicular exchange bias. I assume, although this is not clearly stated by the authors, they wish to use the antiferromagnetic order to store information in a memory element and use the exchange bias to detect the antiferromagnetic Neel vector direction. If so, they need to present the case and significance of the idea more clearly. They should also cite related work that has done readout of electrically switched antiferromagnetic IrMn_3 via exchange bias to a magnetic tunnel junction, which was recently reported in in-plane MTJs and is relevant to this work. [Du et al., Nature Electronics 6, 425 (2023)]

Reply: We are grateful to the referee for the very insightful comment on the importance of device applications in our work and also for the helpful suggestions. As pointed out by the referee, beyond the applications in traditional spintronic devices, the manipulation of the EB effect, intricately associated with the bulk magnetic state of AFM [Schuller et al., *J. Magn. Magn. Mater.*, **416**, 2 (2016)], is expected to be a promising approach for operating data based on high-density information storage with the antiferromagnetic order. The data could potentially be written using the method of full electrical manipulation of perpendicular exchange bias, and be read out employing efficient effects like tunnel magnetoresistance (TMR) [Du. et al., *Nat. Electron.*, 6, 425 (2023)] and tunneling anisotropic magnetoresistance (TAMR) [Wang. et al., *Phys. Rev. Lett.*, 109, 137201 (2012)], which were hoped for application in advanced memory and

logic devices. The application-related motivation of the field-free PEBS was added in the *Introduction* of the revised manuscript.

On the other hand, the main message of our paper is that we explored the underlying physical intension of exchange bias manipulation in a single-crystalline heterostructure with epitaxial strain and provided an efficient method for electrical manipulation of the perpendicular exchange bias effect with the absence of the external magnetic field.

Comment 2: IrMn₃ switching by spin-orbit torque has previously been extensively studied. In addition to the above paper, the authors should also cite and compare their results to this prior work which performs anisotropic magnetoresistance detection. [Arpaci et al., Nature Communications 12, 3828 (2021)]

Reply: Thanks for the suggestion. The paper *Nature Communications* **12**, 3828 (2021) systematically substantiates the switching of the Néel order driven by spin-orbit torque (SOT), presenting comprehensive and reliable evidence. The switching process of the Néel order is effectively detected through the anisotropic magnetoresistance measurements in this work. As discussed in our manuscript, the efforts in electrical manipulation of antiferromagnetic materials primarily focus on two aspects: Néel order in bulk antiferromagnet and the spin configuration at the antiferromagnet/ferromagnet interface, referring to the exchange bias (EB) effect. This article highlights contributions in the former category, and is cited in our revised manuscript. However, in our work, the switching of exchange bias does not originate from the SOT-induced switching of Néel order in the IrMn₃ layer. Instead, it originates from the reorientation of the irreversible part of the uncompensated magnetic moments at the antiferromagnet/ferromagnet interface. The current-induced SOT may exist in the Pt/Co/IrMn structure, but it is not the predominant contribution. The reasons are as follows:

The electrical manipulation of the PEBS is not dependent on the direction of the pulsed

current but rather correlates with the magnetization state of Co. To clarify the contribution of SOT in the PEBS process, the magnetization direction of Co was fixed in the “up” state utilizing a magnetic field in the Pt/Co/IrMn structure, and then a positive and a negative pulsed current were injected to change the polarization direction of the spin current, respectively. It was indicated by the AHE curves of the sample that the perpendicular exchange bias effect remains unchanged and consistently exhibits a negative state, irrespective of the direction of the pulsed current, as shown in Fig. R1a and b. Similarly, when the magnetization of Co was fixed in the “down” state, the perpendicular exchange bias remained positive state, as shown in Fig. R1c and d. The results substantiate that in the epitaxial Pt (3 nm)/Co (1 nm)/IrMn (2 nm) sample, the switching of the perpendicular exchange bias behavior does not originate from the SOT-induced switching behavior of Néel order in the antiferromagnetic IrMn₃ layer. Detailed discussion about the role of SOT in the PEBS process has been added to **P4 of the Supporting Information.**

Fig. R1. AHE loops for the Pt/Co/IrMn₃ multilayer after different pulsed currents with the magnetization of the Co layer set as “up” (a, b) and “down” (c, d) states, respectively.

Comment 3: *IrMn₃ is also expected to exhibit anomalous Hall effect in the absence of a ferromagnetic Co layer. The authors should check this by growing Pt/IrMn₃ bilayers without Co for comparison, and check if the AHE and its electrical switching are also observed in this case.*

Reply: The IrMn₃ alloy exhibits two distinct crystalline structures corresponding to two different spin configurations. Specifically, the chemically ordered $L1_2$ -IrMn₃ features a non-collinear antiferromagnetic spin structure, while the chemically disordered γ -IrMn₃ possesses a non-coplanar $3Q$ -type antiferromagnetic spin structure. The previous work [Chen et al., *Phys. Rev. Lett.*, **112**, 017205 (2014)] predicted substantial anomalous Hall conductivity in the antiferromagnetic IrMn₃ layer, which is attributed to the Kagome lattice formed by Mn atoms, and this unusual triangular magnetic structure presenting only in the non-collinear antiferromagnetic IrMn₃ with $L1_2$ phase. An anomalous Hall conductivity of $40 \Omega^{-1} \text{ cm}^{-1}$ in $L1_2$ -ordered IrMn₃ at room temperature was reported by Iwaki et al. [*Appl. Phys. Lett.*, **116**, 022408 (2020)]. They also compare anomalous Hall conductivity under different ordering degrees of $L1_2$ -ordered IrMn₃. The results indicate that higher chemical ordering correlates with increased anomalous Hall conductivity in $L1_2$ -ordered IrMn₃, indicating that the chemically disordered γ -IrMn is unable to exhibit anomalous Hall effect.

For the deposition of $L1_2$ -IrMn₃, the growth and annealing temperatures need to achieve approximately 500 °C for the formation of the chemically ordered phase. In contrast, the growth of IrMn₃ in our study was performed at room temperature without any subsequent annealing process, resulting in the formation of the chemically disordered γ -IrMn₃. Therefore, the IrMn₃ layer in our work is not expected to manifest anomalous Hall effects.

Moreover, we prepared a controlled sample with a Pt (3 nm)/IrMn₃ (2 nm) structure. Similar Hall bar devices were fabricated, and the anomalous Hall effects measurements were performed. As suspected, the R-H curve of the sample does not exhibit the

hysteresis behavior, indicating the negligible anomalous Hall effect as shown in Fig. R2, making it unfeasible to detect the current switching behavior.

Fig. R2. R-H loop for the Pt (3 nm)/IrMn₃ (2 nm) bilayer.

Comment 4: *The magnitude of the exchange bias in this experiment is surprisingly large, as noted by authors. The paper should provide more information on the number of measured samples and the statistics of the exchange bias, to evaluate the robustness and reproducibility of the result.*

Reply: We summed up the experimental results of the anomalous Hall effect for five batches of the samples with the same structure of Pt (3 nm)/Co (1 nm)/IrMn₃ (2 nm) and extracted the corresponding perpendicular exchange bias field. Samples 1-4 were prepared before writing the manuscript, while sample 5 was deposited recently. Ten devices located at different positions of each sample are involved to confirm the robustness and reproducibility of the considerable perpendicular exchange bias effect as shown in Fig. R3. The distribution of perpendicular exchange bias fields for the fifty devices in the different samples is listed below, ranging from 300 Oe to 500 Oe. Statistical analysis of the perpendicular exchange bias fields for various devices was performed, as shown in Fig. R4. The majority of devices exhibited exchange bias field distributions centered around 400 Oe. Therefore, we conclude that the significant

perpendicular exchange bias effect observed in the Pt (3 nm)/Co (1 nm)/IrMn₃ (2 nm) structure is reproducible and robust. Detailed discussion has been added to the revised manuscript and **P1 of the Supporting Information**.

Fig. R3. H_{PEB} of five different samples extracted from ten devices located at different positions within each sample.

Fig. R4. Statistical analysis of the H_{PEB} for various devices of the five samples.

Comment 5: The authors should also compare to different thicknesses of IrMn₃. For instance, 3 to 4 nm IrMn₃ should have a Neel temperature above room temperature even without the epitaxial strain. Can they also observe perpendicular exchange bias and switch it by current in that case? How large is the exchange bias?

Reply: Samples with thicknesses of 3 nm and 4 nm for the IrMn layer were also prepared. The anomalous Hall curves reveal increased perpendicular exchange bias fields, measured as 811 Oe and 842 Oe, respectively. Furthermore, PEBS measurements on both samples demonstrate the capability for the switching of perpendicular exchange bias. Nevertheless, the sample with 2 nm-thick IrMn₃ is particularly noteworthy owing to the ultrathin antiferromagnetic layer.

In addition, the samples with IrMn thicknesses of 3 nm and 4 nm can still manifest PEBS behavior. This is attributed to the fact that the epitaxial strain at the interface is retained in these two samples with the single-crystalline structure. Furthermore, we conducted experiments on the electrical manipulation of PEBS behavior in a polycrystalline Pt/Co/IrMn system featuring minimal epitaxial strain.

Fig. R5. AHE loops for the Pt/Co/IrMn₃ multilayer with polycrystalline structure after pulsed currents ($3.0 \times 10^7 \text{ A cm}^{-2}$) with different magnetized states of the Co layer, respectively.

As shown in Figure S3 of Supporting Information, the polycrystalline sample with an IrMn thickness of 2 nm didn't exhibit an exchange bias effect. A sample with the

polycrystalline structure of Si (SiO₂)/Pt (3 nm)/Co (1 nm)/IrMn₃ (5 nm) was deposited with a thicker IrMn layer. The AHE curves revealed an exchange bias field of 752 Oe, as depicted in Fig. R5a. By performing similar PEBS operations with a current density of 3.0×10^7 A cm⁻² (which was 2.7×10^7 A cm⁻² for the single-crystalline sample), the exchange bias remained essentially unchanged in the polycrystalline sample. As shown in Fig. R5b and c, the perpendicular exchange bias fields were extracted to be 695 Oe and 697 Oe, respectively. This suggests that the epitaxial strain in the single-crystalline structure is necessary for the field-free PEBS process in heterostructures with ultrathin IrMn Layer.

Reviewer #2

There are a few problems I think that need to be addressed prior to publication. I have marked up a manuscript that I will attach but here are my main points:

Reply: Thanks for the very careful reading of our manuscript. We've carefully incorporated your suggestions, and we believe these revisions have made our work more compelling.

Comment 1: *Line 139 onwards: you talk about the RHEED analysis from different elements in the film. I am assuming that you could distinguish between the different elements in the film by performing the RHEED at different parts of the growth process. This should be made a bit clearer.*

Reply: The capture of RHEED patterns was carried out at different deposition stages corresponding to each layer. With the grazing incidence, the penetration depth of the electron beam is limited to just a few monolayers. This allowed us to monitor the epitaxial growth of each layer individually during the deposition. We appreciate your suggestion, and we have added this clarification in the manuscript.

Comment 2: *Line 189/190: you state that as the M_{Co} rotates around the M_{Mn} rotates with it increasing the coercivity. This is completely true but for clarity it should be stated that there would be no exchange bias which would align better with the rest of the description. This is implied before this statement but it should be made clear.*

Reply: Thanks for the suggestion. Due to the significantly diminished volume anisotropy energy in the ultrathin IrMn layer, the sample should only exhibit an increase in coercivity without the exchange bias effect. We have addressed this point in the manuscript.

Comment 3: Paragraph starting at line 239: there is an explanation of the film Pt/Co/IrMn₃ and then there is reference to Fig. 3. However, in the in Fig 3d there are two more layers MgO/CrO that are not mentioned in the text. Further explanation is needed here.

Reply: In this work, all the samples prepared for electrical transport measurements were capped with a 2nm-thick MgO and 2nm-thick Cr capping layer to prevent oxidation, as described in the *Materials and Methods section*. However, a sample for PNR measurements usually needs to be relatively thicker to achieve a high signal-to-noise ratio for reflectivities curves with respect to wave vector transfer. Therefore, the thickness of the capping Cr layer was increased to 5nm to enhance the fitting accuracy of the data without compromising the properties of the Pt/Co/IrMn core functional layer.

Additionally, the detected CrO_x as shown in the nuclear scattering-length density (nSLD) curve (Figure 3d) is a consequence of the natural oxidation on the Cr capping layer when the sample is exposed to the atmosphere. The CrO_x is thinner than 2 nm, which is separated from the Pt/Co/IrMn stack by the MgO and Cr capping layer, having no impact on the core functional layers. The detailed explanation has been addressed in the manuscript.

Comment 4: PNR Results: Perhaps I have misunderstood something here but polarised neutron reflectometry would not be sensitive to the changes in magnetisation perpendicular to the interfaces in the film. Since your films are mostly magnetised in the perpendicular direction I am not sure how your analysis can work. Perhaps there is sufficient component of magnetisation parallel to the film for this to work. Could you please comment on this please.

Reply: As mentioned by the referee, polarized neutron reflectometry is only sensitive to the changes of magnetization in-plane. In our experiments, a magnetic field of 1.6T was applied to align all magnetic moments in plane during PNR measurements, as

mentioned in the *Materials and Methods section* of the manuscript. The PNR results gave evidence of the global uncompensated magnetic moments in the antiferromagnetic IrMn layer, antiferromagnetic coupling with the magnetic moment of Co. As proved by the Hall loop shown in Figure 1g of the manuscript, the sample with the structure of Pt/Co/IrMn exhibits excellent perpendicular magnetic anisotropy (PMA). Due to the antiferromagnetic coupling between the uncompensated magnetic moment of IrMn and the magnetic moment of Co, it can be concluded that both of them are predominantly oriented perpendicular to the film.

Comment 5: The most important part of this work is the electric manipulation of the exchange bias. Full electrical current control is done by inserting a Cr layer. The ideal thickness is 0.2nm which is beautifully demonstrated with a wedge going along several devices. There doesn't seem to be any explanation of how this works. Why did you insert the Cr layer? Why Cr? Would this work with other elements? Further explanation is required here.

Reply: For the role of Cr wedge in the PEBS process, two potential mechanisms are considered. The introduction of a wedged structure may cause lateral inversion symmetry breaking, achieving field-free magnetization switching as reported in previous work (refer to ref. 63 in the manuscript). The other configuration is to introduce an in-plane component of the exchange bias field (refer to ref. 64 in the manuscript), which can serve as an alternative to the external magnetic fields, facilitating field-free magnetization switching.

Furthermore, experimental validation of these hypotheses was conducted. For device *D* with 0.2 nm-thick Cr insertion, the current-induced field-free magnetization switching could be realized with the current injected perpendicular to the wedge direction. Subsequent tests were performed to evaluate the current-induced magnetization switching behavior with the pulsed current injected along the wedge direction. The field-free magnetization switching behavior can also be observed as shown in Fig. R6.

The results suggest that the field-free switching in this structure is not attributed to the introduction of a wedged structure.

Fig. R6. The current-induced magnetization switching at zero field of device *D* with the pulsed current injected along the wedge direction.

To verify the origin of field-free magnetization switching behavior, we prepared a Pt (3 nm)/Cr (0.2 nm)/Co (1 nm)/IrMn (2 nm) sample without wedged structure and measured magnetic hysteresis loops along both in-plane ($H // x$, which is along $[11\bar{2}]$ direction; $H // y$, which is along $[1\bar{1}0]$ direction) and out-of-plane directions ($H // z$, which is along $[111]$ direction). With the magnetic field perpendicular to the film, a perpendicular exchange bias field of 420 Oe was identified as shown in Fig. R7a, consistent with the result of anomalous Hall effect observed in device *D* as shown in the manuscript. In addition, an exchange bias field of about 50 Oe exists along both the in-plane directions, as shown in Fig. R7b and c, which provide in-plane effective fields during SOT-driven magnetization switching processes, thereby facilitating full electrical manipulation of the perpendicular exchange bias. The exchange bias fields in both the x and y directions align with the results presented in Fig. R6, exhibiting SOT-driven field-free magnetization switching with pulsed current injected along and perpendicular to the wedge direction.

Therefore, in our investigation, the introduction of a 0.2 nm-thick Cr layer resulted in an in-plane component of the exchange bias effect. Consequently, it facilitated field-free magnetization switching driven by SOT, thus enabling full electrical manipulation of the PEBS process.

Fig. R7. M-H loops for the sample with the structure of Pt (3 nm)/Cr (0.2 nm)/Co (1 nm)/IrMn (2 nm) with H along different directions.

The reasons for the introduction of Cr as an ultra-thin insertion:

(1) The introduction of a Cr insertion does not change the epitaxial structure or the (111)-orientation of the Co and the IrMn layer. The RHEED pattern of the Co layer in the sample with Cr insertion as shown in Fig. R8 suggests the same oriented single-crystalline structure with high epitaxial quality as that of the sample without Cr.

Fig. R8. RHEED pattern of the Co layer for the sample with the structure of Pt (3 nm)/Cr (0.2 nm)/Co (1 nm)/IrMn (2 nm). The incident electron beam was parallel to the $[11\bar{2}0]$ direction of the Al_2O_3 (0001) substrate.

(2) The spin diffusion length of Cr is 13.3 nm [Du et al., *Phys. Rev. B*, **90**, 140407(R) (2014)]. Therefore, the 0.2 nm-thick insertion cannot hinder the spin current generated in Pt injecting into the Co layer.

(3) The anomalous Hall curves in Figure 4b of the manuscript demonstrate that the PMA of the Co layer is extremely sensitive to the thickness of the Cr insertion.

Therefore, Cr stands out as an exceptionally fitting choice. Theoretically, elements exhibiting the aforementioned three characteristics may yield similar outcomes, although additional experimental verification is required. The detailed description has been included in the manuscript and **P6 of the Supporting Information**.

Reviewer #3

Comment 1: *The authors acknowledge the nonnegligible thermal effect during the switching process of the PEB. However, the SOT-induced exchange bias switching behavior also exists in systems such as IrMn/CoFeB, IrMn/CoTb, IrMn/NiFe et al. Is the spin current generated from the Pt layer or the IrMn layer important to the perpendicular exchange bias switching process in this work?*

Reply: As mentioned by the referee, the current induced switching of exchange bias driven by SOT has been reported in the multilayers including IrMn. For the Pt (3 nm)/Co (1 nm)/IrMn (2 nm) samples, the spin current generated in the Pt layer was injected into both the Co layer and IrMn layer (considering the spin diffusion length of Co is more than 10 nm [Vila. et al., Phys. Rev. B., **98**, 174414 (2018)]). Therefore, the SOT-induced switching of the Co magnetization could be realized with the pulsed current. However, the switching of perpendicular exchange bias is not predominantly contributed by the spin current in our work. The first reason is that the exchange bias effect is independent of the direction of the spin current as shown in Fig. R1. Moreover, a sample with a structure of Ir (3 nm)/Co (1 nm)/IrMn₃ (2 nm) was prepared with a similar epitaxial structure, perpendicular magnetic anisotropy, and perpendicular exchange bias effect. The spin Hall angle of Ir is approximately 25% of that of Pt [Fache et al., Phys. Rev. B, **102**, 064425 (2022)], which can result in a reduced spin current during the PEBS process. As shown in Fig. R9a and b, a switching behavior of perpendicular exchange bias is still observed by applying the same pulsed current with a density of 2.7×10^{-7} A cm⁻². This suggests that the SOT induced by the spin current is not the predominant factor in the PEBS process in this work.

Fig. R9. AHE loops for the Ir/Co/IrMn₃ multilayer after pulsed currents with the magnetization of the Co layer set as "up" (a) and "down" (b) states, respectively.

Nevertheless, we do not rule out the potential decoupling effect of the spin current in the switching process of perpendicular exchange bias as reported by Du. et al. [Du. et al., *Nat. Electron.*, **6**, 425 (2023)]. It is conceivable that the spin current might decouple the interfacial magnetizations between Co and IrMn, thereby reducing the energy barrier for switching the perpendicular exchange bias.

Comment 2: Figure 1g demonstrates a perpendicular exchange bias field of 436 Oe with a 2 nm-thick IrMn layer. Is this thickness critical for the observed exchange bias effect? Can it be further reduced to a thinner thickness?

Reply: For the ultrathin IrMn Layer, the thickness of 2 nm is critical for the considerable perpendicular exchange bias effect. To prove this, a sample with the 1.5 nm-thick IrMn layer was prepared. The anomalous Hall effect was investigated as shown in Fig. R10, which only exhibited an increased coercivity without manifesting the perpendicular exchange bias effect. Therefore, to preserve the exchange bias effect in the single crystalline Pt/Co/IrMn structure, the critical thickness of the IrMn layer is 2 nm.

Fig. R10. AHE loop for the Pt/Co/IrMn₃ multilayer with the IrMn thickness of 1.5 nm.

Comment 3: An obvious distinction exists in the switching process of the PEB between Fig. S5 and Fig. S11. It involves an overall shift of the AHE loops towards the negative exchange bias state in Fig. S5, but a double-biased state in Fig. S11. What causes this difference?

Reply: In the PEBS process illustrated in Fig. S5, we initially align the magnetization of Co in the perpendicular direction by applying a magnetic field. Subsequently, a pulsed current can effectively switch the perpendicular exchange bias, resulting in the

loop shifting either toward the positive or negative direction. Conversely, in the field-free PEBS process depicted in Fig. S11, the magnetization of Co and the perpendicular exchange bias are simultaneously influenced by the pulsed current. With an appropriate pulsed current, the magnetization of Co is switched partially, resulting in two kinds of magnetic domains of Co corresponding to opposite orientations. Concurrently, the direction of the exchange bias effect also exhibits two directions, corresponding to the different magnetic domains. Therefore, this scenario is characterized by an R-H loop with a double-biased state. In summary, the direction of the exchange bias is dependent on the magnetization direction of Co when the pulsed current is applied.

***Comment 4:** According to the EDS mapping (Fig. 1f), Iridium seems to exist in both the IrMn layer and the Pt layer. Is this caused by elemental diffusion or other factors?*

Reply: For the EDS mapping characterization in such a thin film, the total quantity of Ir atoms in the IrMn₃ alloy is considerably less than that of Pt. Moreover, the M-line of Ir and Pt are very close in EDS (Ir M_{α1} 1979.9 eV, Pt M_{α1} 2050.5 eV, Ir M_β 2053.5 eV, Pt M_β 2127.3 eV), leading to the misinterpretation of Pt atoms in the Pt layer as Ir atoms [José et al., *Appl. Catal. A-Gen.*, **437**, 155 (2012); Maehata et al., *J. Low. Temp. Phys.*, **184**, 5 (2016)]. Additionally, it should be noted that only the IrMn layer and Pt layer exhibit an elemental distribution of Ir based on EDS mapping, while the intermediate Co layer does not. This confirms that the presence of Ir atoms in the Pt layer in EDS mapping is not due to elemental diffusion.

***Comment 5:** Figure 4 presents a series of devices with different Cr thicknesses. Can the full electrical manipulation of the exchange bias process be achieved in these other devices?*

Reply: As mentioned in the manuscript, an innovative strategy to achieve full electrical manipulation of perpendicular exchange bias entails the conception of a sample with the feasibility of field-free SOT switching of Co magnetization. For devices with Cr

thicknesses of 0.25 and 0.3 nm (devices *E* and *F*), the PMA is not good enough, thereby hindering the switching of perpendicular exchange bias. In the case of devices with thinner Cr insertion (devices *A*, *B*, and *C*), the SOT-driven magnetization switching of Co shows a lower switching ratio at zero-field compared to that of device *D* as shown in Fig. 4(c) in the manuscript. Upon applying pulsed current, the switching with a 100% ratio of the Co magnetization cannot be achieved. For instance, in the device with a Cr thickness of 0.15 nm (device *C*), the zero-field current-induced magnetization switching ratio is only 75%, as depicted in Fig. R11, which limited the range of the perpendicular exchange bias field manipulated by the pulsed current.

Therefore, device *D* with a 0.2 nm-thickness Cr, of which the magnetization can be completely switched driven by SOT, exhibits the best performance on the PEBS process.

Fig. R11. AHE loop and current-induced magnetization switching at zero magnetic field for the Pt/Cr/Co/IrMn₃ multilayer with the Cr thickness of 0.15 nm.

Comment 6: Some of the important annotations in certain figures are small, hindering readers' understanding of the results, such as Fig. 3b. It is recommended to adjust for better visibility and comprehension.

Reply: Following the suggestion of the referee, we have revised some annotations in Fig. 2b and Fig. 3b to make the figures more presentable. Thanks for pointing that out.

List of changes

According to the referees' suggestions, we have revised the manuscript, the supporting information, and some figures. The revised parts were highlighted in red in the latest version of the manuscript, and the details are as follows:

1. The discussion for the applications of electrical manipulation of the PEB **“Beyond the applications in...in advanced memory and logic devices”** was added to lines 55-59.
2. More discussion of epitaxial strain in the exchange bias effect was added to lines 80-86 as **“Therefore, one of the paramount challenges lies in...exchange bias effect at the antiferromagnet/ferromagnet interface”**.
3. On page 4, **“The MgO (2 nm)/Cr (2 nm) was deposited as capping layers to avoid degradation”** was added to lines 98-99.
4. On page 7, lines 152-155, the distance between the reflection streak and the first-order diffraction streak was defined as **“ d ”**, which was **“ d_{Pt} ”** and **“ d_{IrMn} ”** for that of Pt layer and IrMn layer, respectively, and the corresponding annotations have been added to **Figure 2b** of the manuscript.
5. On page 9, **“only resulting in an increase in...to produce exchange bias accompanied by an increased coercivity”** was added to lines 198-200.
6. A more detailed description of n_{re} and n_{ir} was added to lines 231-236.
7. The method of extracting n_{re} and n_{ir} was revised in lines 238-242.
8. The mechanism of the field-free magnetization switching driven by SOT for device D was added to lines 304-306.
9. On page 15, the last sentence was changed to **“The study will shed light on...spintronic memory and logic devices”** at lines 326-331.
10. Some annotations in **Figure 3b** were adjusted for better visibility and comprehension.
11. Some grammar mistakes have been corrected in the revised version.
12. The “textured” at line 345 was changed to **“polycrystalline”**.

13. On page 16, **“For the sample prepared for...of data fitting with an increased thickness”** was added to lines 350-353.
14. The references for [25], [26], [27], [35], and [64] were added to the manuscript.
15. Three parts of supporting information were added to the supporting information, consisting of **P1 “Robustness and reproducibility of the PEB with 2-nm IrMn layer”, P4 “The role of SOT in the PEBS process”, and P6 “Field-free SOT switching in Pt/Cr/Co/IrMn”**.

REVIEWER COMMENTS

Reviewer #1 (Remarks to the Author):

I would like to thank the authors for the response to the previous comments, and the new data added in the revision. However, I am still not convinced that this paper is suitable for publication, and some of the new data increases my concerns.

1. The authors show 300 to 500 Oe exchange bias for 2 nm IrMn3 films, but completely zero exchange bias once the IrMn3 thickness is reduced to 1.5 nm. This raises some questions. Why is the exchange bias disappearing so abruptly, and where is the transition thickness? Can the authors identify this thickness, for example, by growing samples with a thickness gradient and studying the exchange bias as a function of position?
2. The authors make a distinction between switching bulk IrMn3 Neel order by SOT, and switching only the interfacial layer that is in contact with Co. This raises two questions. First, since the Co is perpendicular, as is the exchange bias, what is the symmetry breaking mechanism to make deterministic switching of the out of plane magnetization in Co (and thus the exchange bias) by in-plane current possible?
3. Second, what is the evidence that the switching of IrMn3 is confined only to its surface layer?

Reviewer #2 (Remarks to the Author):

This is an important area of research where the exchange bias can be manipulated using a current. As well as providing important information for applications there is also a strong academic interest.

The authors have been very detailed in the response and addressed all my concerns.

Reviewer #3 (Remarks to the Author):

The authors have provided detailed responses to my comments, which are satisfactory. This is a high-quality paper, and in my opinion, it should be accepted.

Response Letter

We thank all the referees for the comments, following which we further revised our manuscript and Supplementary Information. Below we provide the point-by-point responses to the comments of the referees.

Reviewer #1

I would like to thank the authors for the response to the previous comments, and the new data added in the revision. However, I am still not convinced that this paper is suitable for publication, and some of the new data increases my concerns.

Comment 1-1: *The authors show 300 to 500 Oe exchange bias for 2 nm IrMn₃ films, but completely zero exchange bias once the IrMn₃ thickness is reduced to 1.5 nm. This raises some questions. Why is the exchange bias disappearing so abruptly, and where is the transition thickness? Can the authors identify this thickness, for example, by growing samples with a thickness gradient and studying the exchange bias as a function of position?*

Reply: As the referee mentioned, the perpendicular exchange bias effect disappeared once the thickness of the IrMn layer was reduced to 1.5 nm (about 6 monolayers). Based on the study by L. Frangou *et al.*, which accounts for magnetic phase transitions and finite size-scaling (refer to *ref. 54* in the manuscript), a linear relationship between the Néel temperature (T_N) and the thickness of the IrMn layer (t_{IrMn}) has been discerned when the t_{IrMn} is below 2.7 nm (L. Frangou, *Phys. Rev. Lett.*, **116**, 077203 (2016), *ref. 44* in the manuscript), which suggests that the thickness change of 0.5 nm corresponds to a T_N change of more than 50 K. Furthermore, the change rate slows down when the t_{IrMn} is larger than 2.7 nm (Figure 3(a) in *ref. 44*). Typically, t_{IrMn} needs to be larger than 4 nm to obtain a considerable H_{EB} in multilayers. Within this range of thickness, the T_N exhibits a slow variation with thickness, manifested as a gradual change in the exchange bias field. Benefiting from the epitaxial strain in single-crystalline multilayers, our

work has demonstrated significant exchange bias effects in the sample with merely a 2-nm-thick IrMn layer (8 monolayers). In this situation, considering the finite size effect, the T_N vs t_{IrMn} curve should exhibit a steep slope when t_{IrMn} is smaller than 2 nm, corresponding to a sensitive dependence of the antiferromagnet anisotropic energy on the thickness. Therefore, in our opinion, it is reasonable for the disappearance of the exchange bias effect with the t_{IrMn} changing from 2 nm to 1.5 nm.

Fig. R1. AHE loop for the Pt/Co/IrMn₃ multilayer with 1.75-nm-thick IrMn₃ layer at RT.

Furthermore, the transition thickness of the antiferromagnetic IrMn layer to achieve a perpendicular exchange bias effect at room temperature was identified. A sample with an IrMn thickness of 1.75 nm (about 7 monolayers) was prepared. The R-H loop of the sample is shown in Fig. R1. The extracted H_{PEB} is about 87 Oe at room temperature. Moreover, the dependence of H_{PEB} on t_{IrMn} is shown in Fig. R2, indicating that the perpendicular exchange bias effect emerges with t_{IrMn} larger than 1.5 nm. Consequently, the transition thickness of the antiferromagnetic IrMn layer to achieve a perpendicular exchange bias effect at room temperature is identified to be approximately 1.5 nm.

Fig. R2. The curves of H_{PEB} vs t_{IrMn} at room temperature. The black dashed line marks the saturated H_{PEB} for the Pt/Co/IrMn₃ multilayers.

Fig. R3. AHE loops for the Pt/Co/IrMn₃ multilayer with $t_{\text{IrMn}} = 1.5$ nm at different temperatures ranging from 200 K to 310 K, respectively.

The result can also be verified by the temperature dependence of H_{PEB} of the Pt/Co/IrMn (1.5 nm) multilayer. The field-cooling process to 200 K was performed on the sample with $t_{\text{IrMn}} = 1.5$ nm under a -5000 Oe magnetic field. Then the AHE loops

of the sample with increasing temperatures ranging from 200 K to 310 K were measured as shown in Fig. R3. The extracted H_{PEB} as a function of T was exhibited in Fig. R4. The H_{PEB} decreased with the increasing temperature and was negligible at 300 K, demonstrating that the transition thickness of the antiferromagnetic IrMn layer at room temperature is approximately 1.5 nm. The corresponding results were added to the **P2 of the Supporting Information.**

Fig. R4. The curve of H_{PEB} as a function of T for the sample with $t_{\text{IrMn}} = 1.5$ nm. The black dashed lines mark the blocking temperature for the sample.

Comment 1-2. The authors make a distinction between switching bulk IrMn₃ Néel order by SOT, and switching only the interfacial layer that is in contact with Co. This raises two questions. First, since the Co is perpendicular, as is the exchange bias, what is the symmetry breaking mechanism to make deterministic switching of the out of plane magnetization in Co (and thus the exchange bias) by in-plane current possible?

Reply: To realize the full electrical manipulation of the exchange bias, symmetry breaking is necessary for the SOT-induced deterministic magnetization switching of Co with PMA. In our work, the perpendicular exchange biased system with Pt/Co/IrMn

structure exhibits great PMA, and the SOT switching in the absence of the external in-plane magnetic field can't be realized, as shown in Fig. 1h of the manuscript. In contrast, field-free switching driven by SOT can be achieved in the Pt/Cr/Co/IrMn structure, as shown in Fig. 4c. The symmetry-breaking mechanism of the Pt/Cr/Co/IrMn structure was clarified to be the in-plane components of the exchange bias field, which can be found in the last response letter (reply to comment 5 of the reviewer 2), and the corresponding results were added to **P7 of the Supporting Information**. Below are the detailed results and analyses.

In the context of the symmetry-breaking in the Pt/Cr/Co/IrMn structure, two potential mechanisms are considered. The introduction of a wedged structure may cause lateral inversion symmetry breaking, achieving field-free magnetization switching as reported in previous work (refer to ref. 63 in the manuscript). The other configuration is to introduce an in-plane component of the exchange bias field (refer to ref. 64 in the manuscript), which can serve as an alternative to the external magnetic fields, facilitating field-free magnetization switching. Furthermore, experimental validation of these hypotheses was conducted. As illustrated in Fig. R5a, the x , y , and z axes are defined along $[11\bar{2}]$, $[1\bar{1}0]$, and $[111]$ direction of the Pt layer, respectively. For the device with 0.2-nm-thick Cr insertion, the current-induced field-free magnetization switching could be realized with the current injected perpendicular to the wedge direction ($I // x$). Subsequent tests were performed to evaluate the current-induced magnetization switching behavior with the pulsed current injected along the wedge direction ($I // y$). The field-free magnetization switching behavior can also be observed as shown in Fig. R5c. The results suggest that the field-free switching in this structure is not attributed to the wedged structure. In addition, the effective fields for both configurations were identified by applying a negative in-plane magnetic field, under which the SOT-induced magnetization switching ratio is negligible. As shown in red symbols in Fig. R5b and c, the corresponding effective fields were clarified to be 65 Oe and 70 Oe, respectively.

Figure R5. **a**, Schematic structure of the epitaxial Pt/Cr/Co/IrMn₃ multilayer. The current-induced magnetization switching at zero field (blue symbols) and non-zero field (red symbols) with the pulsed current injected perpendicular to **(b)** and along **(c)** the wedge direction, respectively.

To verify the origin of field-free magnetization switching behavior, a sample without the Cr wedge [Pt (3 nm)/Cr (0.2 nm)/Co (1 nm)/IrMn (2 nm)] was prepared. The magnetic hysteresis loops of the sample were measured along both in-plane ($H // x$; $H // y$) and out-of-plane ($H // z$) directions. With the magnetic field perpendicular to the film, a perpendicular exchange bias field of 420 Oe was identified, consistent with the AHE result observed in device *D* as shown in the manuscript. As shown in Fig. R6a and b, exchange bias fields of about 50 Oe exist along both the in-plane directions, which is comparable to the effective fields in the SOT-induced magnetization switching process (Fig. R5b and c). Consequently, the in-plane exchange bias fields result in symmetry breaking in the SOT-driven magnetization switching processes, thereby facilitating full electrical manipulation of the perpendicular exchange bias.

Fig. R6. M - H loops for the sample with the structure of Pt (3 nm)/Cr (0.2 nm)/Co (1 nm)/IrMn (2 nm) with H along different directions.

Comment 2. Second, what is the evidence that the switching of IrMn₃ is confined only to its surface layer?

Reply: It seems there might be a misunderstanding regarding this point. In our opinion, the PEBS process originates from the switching of the irreversible part of the uncompensated magnetization at the interface of the IrMn layer, rather than the switching of its Néel order. However, it should be noticed that this does not imply that the PEBS process is not accompanied by the switching of the Néel order of the IrMn layer. There might be a spin current, originating from the Pt layer with strong spin-orbit coupling, injected into the IrMn layer during the PEBS process, which potentially leads to the switching of the Néel order. But this does not directly affect the exchange bias effect of the Pt/Co/IrMn. Considering that the exchange bias effect is an interfacial effect originating from the ferromagnet/antiferromagnet interface, the changes in the exchange bias field should correspond to a transition of the interfacial spin configuration of the Mn atoms, which has been proven by XMCD experiments in previous work [H. Ohldag et al., *Phys. Rev. Lett.*, **91** 017203 (2003)].

In summary, although the injection of pulsed current may be accompanied by a switching of the Néel order, which may affect the interfacial spin configuration by bulk spin structure (thus the exchange bias effect), the switching behavior of the irreversible part of the interfacial uncompensated magnetization is the origin of the PEBS process.

Reviewer #2

This is an important area of research where the exchange bias can be manipulated using a current. As well as providing important information for applications there is also a strong academic interest.

The authors have been very detailed in the response and addressed all my concerns.

Reviewer #3

The authors have provided detailed responses to my comments, which are satisfactory.

This is a high-quality paper, and in my opinion, it should be accepted.

Reply: We are grateful to the Reviewer #2 and Reviewer #3 for their recommendation for publication.

REVIEWER COMMENTS

Reviewer #1 (Remarks to the Author):

I would like to thank the authors for the detailed response and additional data, which mostly answer my questions. My only remaining question is about the third point raised in the previous review. The authors clarify that they understand their observations in terms of switching of the uncompensated magnetization at the interface, rather than the Neel vector. It is not clear to me that these two can be decoupled. I recommend that the authors explain this point in more detail.

Response Letter

Reviewer #1

I would like to thank the authors for the detailed response and additional data, which mostly answer my questions. My only remaining question is about the third point raised in the previous review. The authors clarify that they understand their observations in terms of switching of the uncompensated magnetization at the interface, rather than the Néel vector. It is not clear to me that these two can be decoupled. I recommend that the authors explain this point in more detail.

Reply:

Thanks for the suggestion. The switching of PEB corresponds to the reorientation of the irreversible portion of the uncompensated magnetization at the IrMn surface (\mathbf{n}_{ir}). For a stable exchange-biased system, \mathbf{n}_{ir} should be strongly coupled with its bulk spins (which determines the Néel vector, $\mathbf{n}_{\text{Néel}}$), and this is the precondition of the emergence of the exchange bias effect. In contrast, for the current-induced PEBS process, \mathbf{n}_{ir} is coupled with the magnetization of the Co layer (\mathbf{m}_{Co}) in terms of energy, rather than the $\mathbf{n}_{\text{Néel}}$. The reasons are as follows:

As we know, the exchange bias effect is dependent on the competition between the antiferromagnetic volume anisotropy energy (E_A , which can affect the coupling between \mathbf{n}_{ir} and $\mathbf{n}_{\text{Néel}}$) and the exchange energy at the Co/IrMn interface (E_{int} , related to the coupling between \mathbf{n}_{ir} and \mathbf{m}_{Co}). For an exchange-biased system, $E_A > E_{\text{int}}$. The \mathbf{n}_{ir} is strongly coupled with the bulk spins of the IrMn layer, which can not be switched with \mathbf{m}_{Co} , resulting in a unidirectional anisotropy for Co. While in the PEBS process in this work, a thermal accumulation induced by pulsed current exists within the device. The raised temperature results in a marked reduction of E_A ($E_A < E_{\text{int}}$), weakening the coupling between \mathbf{n}_{ir} and $\mathbf{n}_{\text{Néel}}$. Therefore, the \mathbf{n}_{ir} prefers to be coupled with \mathbf{m}_{Co} , rather than $\mathbf{n}_{\text{Néel}}$. As a result, the direction of the \mathbf{n}_{ir} is only determined by \mathbf{m}_{Co} during the PEBS process. After that, the reoriented \mathbf{n}_{ir} is recoupled with $\mathbf{n}_{\text{Néel}}$, achieving the switching of

the exchange bias effect. In summary, during the PEBS process, the weakened E_A results in the decoupling between \mathbf{n}_{Ir} and $\mathbf{n}_{\text{Néel}}$. Then the \mathbf{n}_{Ir} can be reoriented according to the coupling with \mathbf{m}_{Co} , thus exhibiting the exchange bias switching behavior.

On the other hand, the $\mathbf{n}_{\text{Néel}}$ can be driven by SOT, which may also be a potential approach for the manipulation of EB. However, SOT-switching of the $\mathbf{n}_{\text{Néel}}$ is not the dominant role in this work. Indeed, there might be a spin current, originating from the Pt layer with strong spin-orbit coupling, injected into the IrMn layer during the PEBS process. However, our experimental results demonstrate that the direction of the exchange bias effect is only related to the \mathbf{m}_{Co} , independent of the pulsed current direction, as exhibited in **Fig. S10** of the Supporting Information. Therefore, the switching of the perpendicular exchange bias in this work does not correspond to the SOT-induced switching of $\mathbf{n}_{\text{Néel}}$.

REVIEWER COMMENTS

Reviewer #1 (Remarks to the Author):

Thank you for the detailed explanation of the distinct roles of the volume anisotropy energy (E_A) and interfacial exchange energy (E_{int}). I have to say that I am not yet fully convinced by this explanation, which in its present form is only a hypothesis without direct evidence offered by the authors. I find it equally likely that the Neel vector switching by SOT is the mechanism of switching in their samples. Nonetheless, I am willing to recommend this paper for publication if the new explanation (the content of the last response) is incorporated into the main text of their paper, so that readers at least can fully understand the authors' hypothesis and interpretation of their data.

Response Letter

Reviewer #1

Thank you for the detailed explanation of the distinct roles of the volume anisotropy energy (E_A) and interfacial exchange energy (E_{int}). I have to say that I am not yet fully convinced by this explanation, which in its present form is only a hypothesis without direct evidence offered by the authors. I find it equally likely that the Neel vector switching by SOT is the mechanism of switching in their samples. Nonetheless, I am willing to recommend this paper for publication if the new explanation (the content of the last response) is incorporated into the main text of their paper, so that readers at least can fully understand the authors' hypothesis and interpretation of their data.

Reply:

We are grateful to the reviewer for the recommendation of our work. Following the suggestion, we have incorporated a detailed discussion in our revised manuscript (**page 13-14, line 293-line 302**) based on the content of the last response letter to provide an interpretation of the PEBS mechanism, elucidating the theoretical underpinnings.

REVIEWERS' COMMENTS

Reviewer #1 (Remarks to the Author):

The authors have addressed all my comments. I recommend the paper for publication.